# Uncertainty Evaluation Metric for Brain Tumour Segmentation

**Raghav Mehta**                                          RAGHAV@CIM.MCGILL.CA
*Centre for Intelligent Machines, McGill University, Canada*

**Angelos Filos**                                          ANGELOS.FILOS@CS.OX.AC.UK
**Yarin Gal**                                          YARIN.GAL@CS.OX.AC.UK
*Oxford Applied and Theoretical Machine Learning Group, University of Oxford, England*

**Tal Arbel**                                          ARBEL@CIM.MCGILL.CA
*Centre for Intelligent Machines, McGill University, Canada*

## Abstract

In this paper, we develop a metric designed to assess and rank uncertainty measures for the task of brain tumour sub-tissue segmentation in the BraTS 2019 sub-challenge on uncertainty quantification. The metric is designed to: (1) reward uncertainty measures where high confidence is assigned to correct assertions, and where incorrect assertions are assigned low confidence and (2) penalize measures that have higher percentages of under-confident correct assertions. Here, the workings of the components of the metric are explored based on a number of popular uncertainty measures evaluated on the BraTS 2019 dataset.

**Keywords:** Brain Tumour Segmentation, Deep Neural Network, Uncertainty.

## 1. Introduction

Deep Neural Networks (DNN) have shown to outperform traditional machine learning methods on a variety of automatic medical image segmentation tasks (Isensee et al., 2018; Išgum et al., 2015; Roth et al., 2015), including tumour segmentation, as depicted by the highest ranking results on recent BraTS challenges (Bakas et al., 2018). However, errors in brain tumour segmentation deter the adoption of DNN frameworks in clinical contexts, particularly those that rely on high voxel-level accuracy, such as in image-guide neurosurgery. Although popular DNNs (Çiçek et al., 2016; Kamnitsas et al., 2017) for brain tumour segmentation provide the "sigmoid"/"softmax" predictions for tumour labels, it is the overall *model uncertainties* which would more informative in assisting clinicians in making more informed decisions. Although, several recent methods (Gal and Ghahramani, 2016; Lakshminarayanan et al., 2017; Maddox et al., 2019) have been proposed to estimate uncertainties in deep neural networks, there is no established strategy in which their usefulness can be assessed and compared for particular clinical contexts. In this paper, we develop metrics to measure the quality of different uncertainty measures for the task of brain tumour segmentation, with the objectives: (1) when the network is correct it is confident in the predicted labels, and (2) when they are incorrect, it is not confident. These metrics were combined and used as the basis of ranking the uncertainties produced by participating teams in the BraTS 2019 sub-challenge on uncertainty quantification.

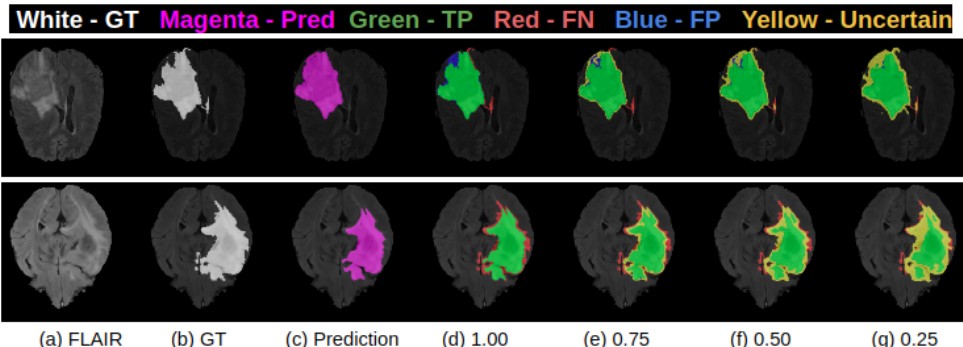

Figure 1: Effect of uncertainty thresholding on two examples for whole tumour segmentations (Top and bottom rows). (a) FLAIR MRI, (b) "Ground truth" labels, (c) Sample prediction, (d) Prediction with no filtering, and (e)-(g) Filtering with uncertainty thresholds ($\tau$) of 0.75, 0.5 and 0.25.

| | Dice | | | |
|---|---|---|---|---|
| | **Dice at 1.00** | **Dice at 0.75** | **Dice at 0.50** | **Dice at 0.25** |
| **Example-1** | 0.94 | 0.96 | 0.965 | 0.97 |
| **Example-2** | 0.92 | 0.955 | 0.97 | 0.975 |
| | **Ratio of Filtered TPs (($TP_{1.00}$ - $TP\tau$) / $TP_{1.00}$)** | | | |
| | **FTP at 1.00** | **FTP at 0.75** | **FTP at 0.50** | **FTP at 0.25** |
| **Example-1** | 0.00 | 0.00 | 0.05 | 0.1 |
| **Example-2** | 0.00 | 0.00 | 0.15 | 0.25 |
| | **Ratio of Filtered TNs (($TN_{1.00}$ - $TN\tau$) / $TN_{1.00}$)** | | | |
| | **FTN at 1.00** | **FTN at 0.75** | **FTN at 0.50** | **FTN at 0.25** |
| **Example-1** | 0.00 | 0.0015 | 0.0016 | 0.0019 |
| **Example-2** | 0.00 | 0.0015 | 0.0026 | 0.0096 |

Table 1: Changes in Dice, Filtered True Positives (FTP), and Filtered True Negatives (FTN) with different uncertainty thresholds ($\tau$) for two different examples.

## 2. Metric for Assessing and Comparing Uncertainty Measures

In the context of the BraTS 2019 challenge, each team provided their multi-class brain tumour segmentation output labels and the voxel-wise uncertainties for each of the associated tasks: whole tumour (WT), tumour core (TC) and enhanced tumour (ET) segmentations. For each task, the uncertain voxels were filtered out at several predetermined (N) number of uncertainty thresholds, $\tau$, and the model performance was assessed based on the contextual metric of interest (here, Dice score) on the remaining voxels at each of the thresholds. For example, at $\tau = 0.75$, all voxels with uncertainty values $\geq 0.75$ are marked as uncertain. The associated predictions are filtered out, and Dice values are calculated for the remaining predictions based on the unfiltered voxels. This evaluation rewards models where the confidence in the incorrect assertions (False Positives - FPs, and False Negatives - FNs) is low and high for correct assertions (True Positives - TPs and True Negatives - TNs). For these models, it is expected that as more uncertain voxels are filtered out, the Dice score should increase on the remaining predictions.

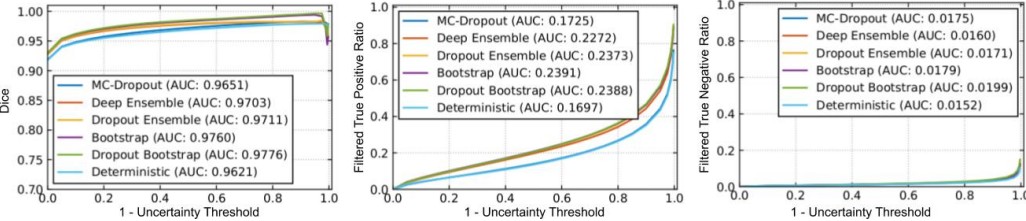

Figure 2: Effect of changing uncertainty threshold ($\tau$) on whole tumour for entropy measure.

The proposed strategy does not keep track of the number of correctly labeled voxels that are filtered at each threshold level along with the uncertain incorrect labels. In order to penalize filtering out many correctly predicted voxels (TPs, TNs) when attaining high Dice values, an additional assessment component is added to keep track of the filtered TP and TNs voxels. Given that tumour segmentation is expected to have a high-class imbalance between tumour and healthy tissues, the system keeps track of the filtered TPs and TNs separately. The ratio of filtered TPs (FTP) at different thresholds ($\tau$) is measured relative to the unfiltered values ($\tau = 1.00$) such that FTP = (TP$_{1.00}$ - TP$_\tau$) / TP$_{1.00}$. The ratio of filtered TNs is calculated in a similar manner. This evaluation essentially penalizes approaches that filter out a large percentage of TP or TN relative to $\tau = 1.00$ voxels in order to attain the reported Dice value.

Figure 1 and Table 1 shows the workings of the assessment metric for example cases based on images from BraTS 2019. Decreasing the threshold ($\tau$) leads to filtering out voxels with incorrect assertions, leading to an increase in the Dice value for the remaining voxels. Case 2 shows a marginally higher Dice value than Case 1 at uncertainty thresholds $\tau = 0.50$ and 0.25. However, the Ratio of Filtered TPs and TNs indicates that this is at the expense of marking more TPs and TNs as uncertain.

Finally, different uncertainty measures are ranked according to a unified score which combines the area under three curves: 1) Dice vs $\tau$, 2) FTP vs $\tau$, and 3) FTN vs $\tau$, for different values of $\tau$. The unified score is calculated as follows:

$$score = \frac{AUC_1 + (1 - AUC_2) + (1 - AUC_3)}{3}. \tag{1}$$

## 3. Experiments and Results

A modified 3D U-Net architecture (Çiçek et al., 2016; Mehta and Arbel, 2018) generates the segmentation outputs and corresponding uncertainties. We train (228), validate (57), and test (50) this network based on the publicly available Brain Tumour Segmentation (BraTS) challenge 2019 training dataset (335) (Bakas et al., 2018). The performances of whole tumour segmentation with the Entropy uncertainty measure (Gal et al., 2017), which captures the average amount of information contained in the predictive distribution, using MC-Dropout (Gal and Ghahramani, 2016), Deep Ensemble (Lakshminarayanan et al., 2017), Dropout Ensemble (Smith and Gal, 2018), Bootstrap, Dropout Bootstrap, and Deterministic, are shown in Figure 2. [1] Dropout bootstrap shows the best Dice performance

---

1. Please refer to supplementary material for more results.

(highest AUC), but also has the worst performance for Filtered True Positive and Filtered True Negative curves (highest AUC). This result shows that, here, the higher performance in Dice is at the expense of a higher number of filtered correct voxels. Overall, the metric is working in line with the objectives. However, there is no clear winner amongst these uncertainty methods in terms of rankings.

## 4. Conclusion

This paper provides a rationale behind the design of a metric presented at the MICCAI BraTS 2019 sub-challenge to evaluate and rank uncertainties produced by different methods for brain tumour segmentation. Using two different examples it was demonstrated that the designed metric is able to reward methods which convey higher uncertainty for incorrect assertions and penalize methods which have higher uncertainties for correct assertions.

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

## Supplementary Material

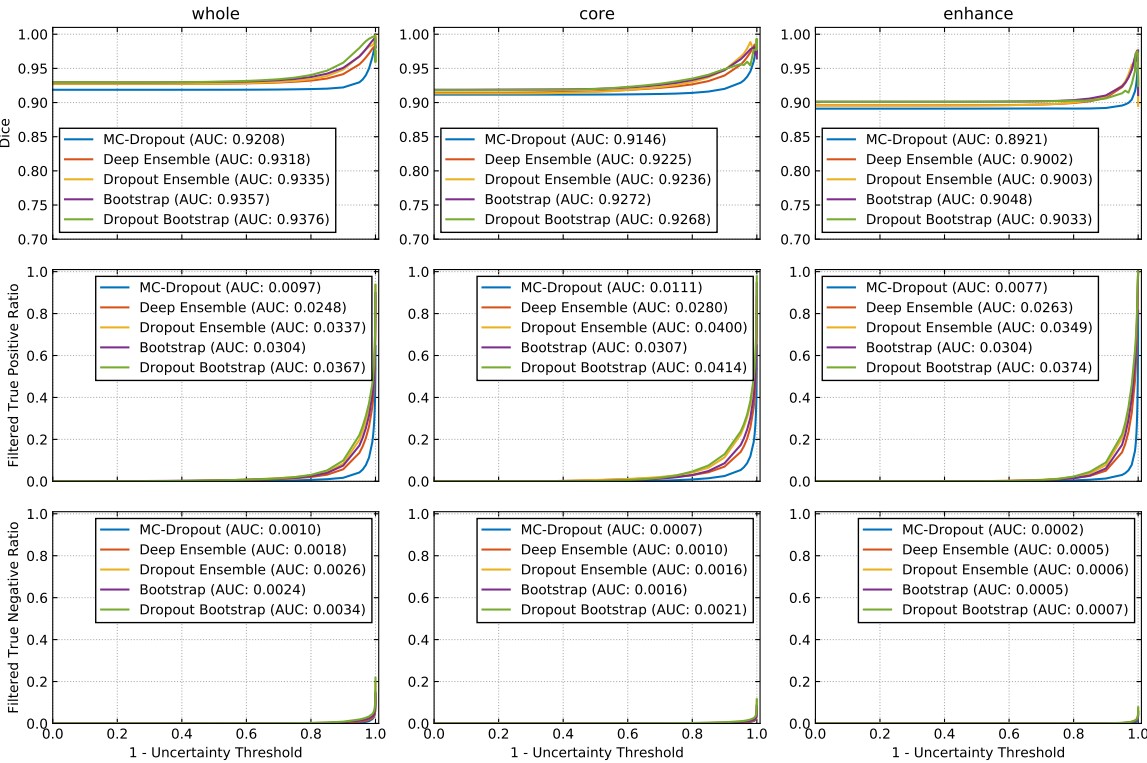

Figure S1: Effect of changing uncertainty thresholds on 3 different tumour sub-types (whole tumour, core tumour, and enhance tumour) for the MI uncertainty measure.

|  | Whole Tumour | Tumour Core | Enhancing Tumour |
|---|---|---|---|
| **MC-Dropout** | 0.9700 | 0.9676 | 0.9614 |
| **Deep Ensemble** | 0.9684 | 0.9645 | 0.9578 |
| **Dropout Ensemble** | 0.9657 | 0.9607 | 0.9549 |
| **Bootstrap** | 0.9676 | 0.9650 | 0.9580 |
| **Dropout Bootstrap** | 0.9658 | 0.9611 | 0.9551 |

Table S1: An example of the resulting final score (Eq.1) for three different tumour types for the MI uncertainty measure.

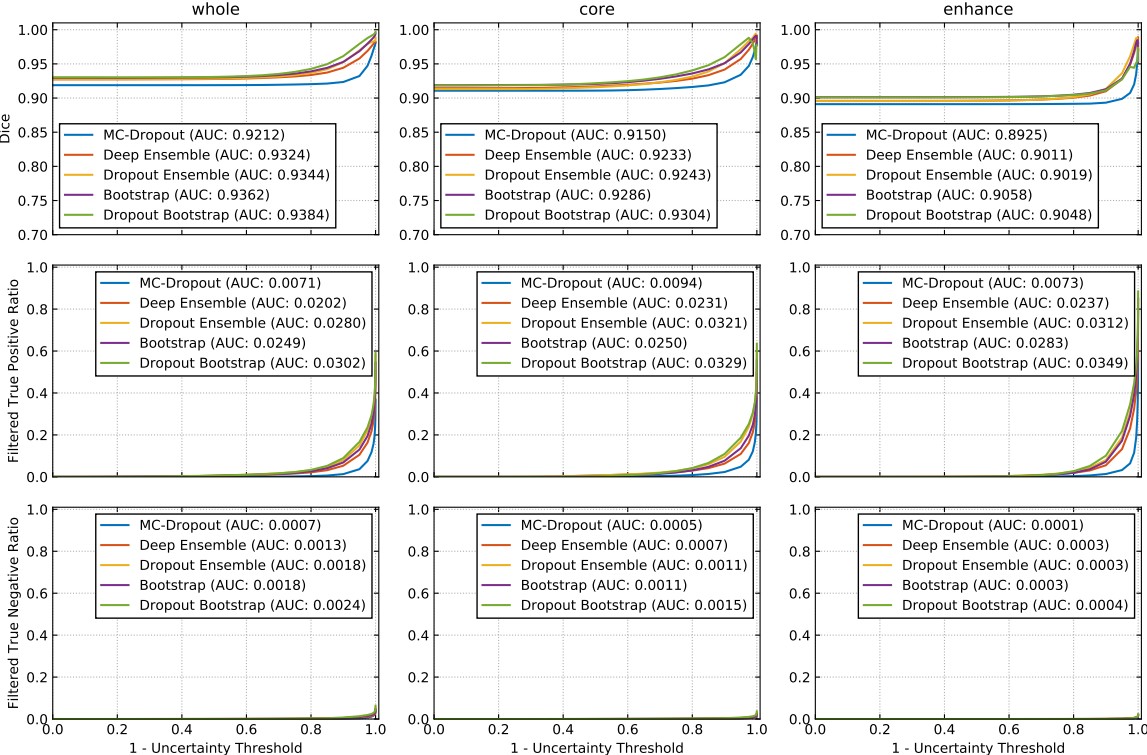

Figure S2:  Effect of changing uncertainty thresholds on 3 different tumour sub-types (whole tumour, core tumour, and enhance tumour) for the sample variance uncertainty measure.

|  | Whole Tumour | Tumour Core | Enhancing Tumour |
|---|---|---|---|
| **MC-Dropout** | 0.9711 | 0.9684 | 0.9617 |
| **Deep Ensemble** | 0.9703 | 0.9665 | 0.9590 |
| **Dropout Ensemble** | 0.9682 | 0.9637 | 0.9568 |
| **Bootstrap** | 0.9698 | 0.9675 | 0.9591 |
| **Dropout Bootstrap** | 0.9686 | 0.9653 | 0.9565 |

Table S2:  An example of the resulting final score (Eq.1) for three different tumour types for the sample variance uncertainty measure.

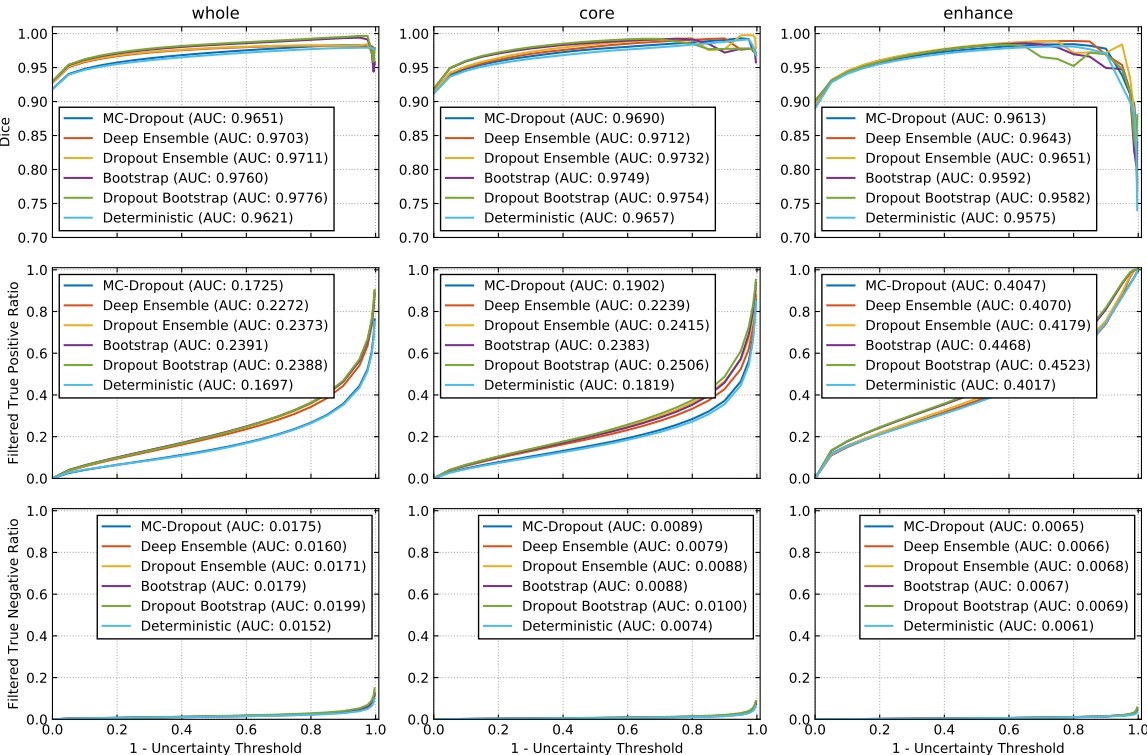

Figure S3: Effect of changing uncertainty thresholds on 3 different tumour sub-types (whole tumour, core tumour, and enhance tumour) for the entropy uncertainty measure.

|  | Whole Tumour | Tumour Core | Enhancing Tumour |
|---|---|---|---|
| **MC-Dropout** | 0.9250 | 0.9233 | 0.8500 |
| **Deep Ensemble** | 0.9090 | 0.9131 | 0.8502 |
| **Dropout Ensemble** | 0.9056 | 0.9076 | 0.8468 |
| **Bootstrap** | 0.9063 | 0.9093 | 0.8352 |
| **Dropout Bootstrap** | 0.9063 | 0.9049 | 0.8330 |
| **Determinstic** | 0.9257 | 0.9255 | 0.8499 |

Table S3: An example of the resulting final score (Eq.1) for three different tumour types for the entropy uncertainty measure.

