# OpenReview forum: "Uncertainty Evaluation Metric for Brain Tumour Segmentation"
_MIDL.io/2020/Conference — MIDL 2020_

### Official Review · AnonReviewer2 · 2020-03-10
**Valuable, well-written contribution with a few shortcomings**

**Rating:** 3
**Confidence:** 5

**Review:**

Quality and clarity:
- The short paper is well-written and easy to follow.

Significance:
- The evaluation of uncertainty estimations in segmentation is crucial. Given the amount of existing uncertainty estimation methods, such metrics are critical to compare the produced uncertainty estimates quantitatively.

Pros:
- The work addresses an important problem.
- The proposed metric not only rewards uncertainties in the FP and FN but also penalizes the uncertainties in the TP and TN regions.
- Figure 1 and Table 1 greatly improve the understanding of the proposed metric.

Cons:
- The proposed metric is rather complicated to interpret since it consists of three sub-metrics and requires different thresholds.
- The work neither describes how to combine the three sub-metrics, nor it explains how to combine the values at each threshold. Being able to summarize the metric into one scalar value would be beneficial for broader adoption and better interpretation.
- The compared uncertainty estimation methods are insufficiently described or cited.

Minor:
- Typo in Table 1: The TP in the definition of the FTN should probably be a TN.
- The work mentions inter-rater variability as ground truth uncertainty. It is arguable if the desired uncertainty of a model should be similar/identical to the inter-rater disagreement.

---

### Official Review · AnonReviewer4 · 2020-03-14
**The authors in this paper propose a metric to assess uncertainty measures in case of brain tumor segmentation from MRI images.**

**Rating:** 3
**Confidence:** 2

**Review:**

- How do these authors envision this approach will be used in clinical practice? How will a radiologist interact with such system outputs that provide uncertainty estimates?
- Please provide more information on the modified 3D UNET utilized. Did any of the parameters change during experimentation?
- Additional details with respect to the experimentation must be added.
- Additional examples capturing the effectiveness of this metric must be provides.

---

### Official Review · AnonReviewer3 · 2020-03-19
**Interesting to see such efforts in a short paper!**

**Rating:** 3
**Confidence:** 5

**Review:**

The paper presents an evaluation of recently developed uncertainty measures on Brain Tumour Segmentation.

Pros: The paper is well-written and relevant to MIDL topics. Further, it introduces two additional metrics to evaluate the performance of uncertainty on a publicly available database.

Cons:  Calibration wasn't performed and discussed here. The paper would have been even stronger if a quantitative assessment against labels uncertainty, due to intra/inter-observer variability, was performed.

Detailed Feedback:
- As you might know, predictive uncertainty is underestimated, and calibration has been recently investigated in this context, e.g. Guo et al. [1]. Some methods claimed better calibration, e.g. Deep Ensemble. So i was wondering whether reported uncertainty methods were well-calibrated on a validation set or not. It would have been better if the methods were well-calibrated first before running the evaluation, or at least the authors have discussed this point in this discussion and conclusion.
- One of the concluding remarks that I was hoping to see is the need of novel techniques and tools that measure the labels uncertainty, similar to the work of Tomczack et al. [2]. I think this is extremely important as we need to urge researchers to look at this.

[1] Guo, C., Pleiss, G., Sun, Y. and Weinberger, K.Q., 2017, August. On calibration of modern neural networks. In Proceedings of the 34th International Conference on Machine Learning-Volume 70 (pp. 1321-1330). JMLR. org.

[2] Tomczack, A., Navab, N. and Albarqouni, S., 2019. Learn to estimate labels uncertainty for quality assurance. arXiv preprint arXiv:1909.08058.

---

### Official Review · AnonReviewer1 · 2020-03-20
**Novel metrics are proposed to evaluate uncertainty estimation and applied to brain tumor segmentaiton**

**Rating:** 3
**Confidence:** 4

**Review:**

The authors propose some metrics based on thresholded uncertainty to evaluate the reliability of uncertainty estimation methods for deep learning-based segmentation, which is of interest to the community. Effect of these metrics on brain tumor segmentation has been shown. However, the proposed metrics failed to rank different uncertainty estimation methods as in the results.

pros:
1, considering the ratio of filtered TPs and TNs is a reasonable idea for uncertainty assessment.
2, the authors showed some results with a brain tumor segmentation task, which helped to understand the proposed metrics.

cons:
1, using Dice based on thresholded uncertainty to evaluate the uncertainty estimation method has been proposed before, such as the following paper:

[1] Assessing Reliability and Challenges of Uncertainty Estimations for Medical Image Segmentation, MICCAI 2019.

Authors in [1] found that based on such a metric, model ensemble had a better performance than other uncertainty estimation methods. But this paper found that there was no obvious winner  among different uncertainty estimation methods according to the metrics used in this paper. Could the authors explain more about this?

2, following the above problem, the results didn't show the proposed metrics have the ability to distinguish good and poor uncertainty estimation methods. How to validate the effectiveness of the proposed metrics?

---

### Meta-Review · Area_Chair1 · 2020-04-06
**MetaReview of Paper282 by AreaChair1**

**Rating:** 3

**Metareview:**

This paper presents a simple yet effective method to evaluate uncertainty applied to tumor segmentation problem.

This short paper is well written and the results seem relevant to MIDL.



**Paper Type:**

methodological development

---

### Decision · Program_Chairs · 2020-04-11

Accept